# Transcriptome Analysis Revealed ZmPTOX1 Is Required for Seedling Development and Stress Tolerance in Maize

**DOI:** 10.3390/plants13172346

**Published:** 2024-08-23

**Authors:** Yixuan Peng, Zhi Liang, Xindong Qing, Motong Wen, Zhipeng Yuan, Quanquan Chen, Xuemei Du, Riliang Gu, Jianhua Wang, Li Li

**Affiliations:** 1Sanya Institute, China Agricultural University, Sanya 572025, China; 18811716268@163.com (Y.P.); liangzhinzz@163.com (Z.L.); qxd@cau.edu.cn (X.Q.); wenmotong990205@163.com (M.W.); iuenzipo@163.com (Z.Y.); chenquanquan@cau.edu.cn (Q.C.); duxuemei@cau.edu.cn (X.D.); rilianggu@cau.edu.cn (R.G.); wangjh63@cau.edu.cn (J.W.); 2State Key Laboratory of Maize Bio-Breeding, Key Laboratory of Crop Heterosis Utilization, Ministry of Education, Beijing Innovation Center for Crop Seed Technology (MOA), College of Agronomy and Biotechnology, China Agricultural University, Beijing 100193, China; 3Key Laboratory of Cultivation and Utilization of Oil Tea Resources of Jiangxi Province, Jiangxi Academy Forestry, Nanchang 330013, China

**Keywords:** zebra leaf, photosynthesis, circadian rhythm, stress resistance, reactive oxygen species

## Abstract

Plant seedling morphogenesis is considerably related to photosynthesis, pigment synthesis, and circadian periodicity during seedling development. We identified and cloned a maize zebra or crossbanding leaves mutant *wk3735*, which produces pale white kernels and was identified and plays a role in the equilibrium of the Redox state the in/out of ETC by active oxygen scavenging. Interestingly, it produces the zebra leaves during the production of the first seven leaves, which is apparently different from the mutation of homologs AtPTOX in *Arabidopsis*. It is intriguing to investigate how and why yellow crossbands (zebra leaf phenotype) emerge on leaves. As expected, chlorophyll concentration and photosynthetic efficiency both significantly declined in the yellow sector of *wk3735* leaves. Meanwhile, we observed the circadian expression pattern of *ZmPTOX1*, which was further validated by protein interaction assays of the circadian clock protein TIM1 and ZmPTOX1. The transcriptome data of yellow (muW) and green (muG) sectors of knock-out lines and normal leaves of overexpression lines (OE) at the 5th-leaf seedling stage were analyzed. Zebra leaf etiolated sections exhibit a marked defect in the expression of genes involved in the circadian rhythm and rhythmic stress (light and cold stress) responses than green sections. According to the analysis of co-DEGs of muW vs. OE and muG vs. OE, terms linked to cell repair function were upregulated while those linked to environmental adaptability and stress response were downregulated due to the mutation of *ZmPTOX1*. Further gene expression level analyses of reactive oxygen species (ROS) scavenging enzymes and detection of ROS deposition indicated that *ZmPTOX1* played an essential role in plant stress resistance and ROS homeostasis. The pleiotropic roles of ZmPTOX1 in plant ROS homeostasis maintenance, stress response, and circadian rhythm character may collectively explain the phenotype of zebra leaves during *wk3735* seedling development.

## 1. Introduction

The leaf is an important organ for photosynthesis and gas exchange in plants [1]. Plant leaves are colored due to a range of pigments found in the leaves, which are impacted by both internal and external causes. The three basic types of pigment that influence the color of leaves in higher plants are chlorophyll, carotenoids, and flavonoids [2]. Carotenoids are lipid-soluble pigments that are incorporated in chlorophyll and chromoplast membranes, and they can prevent the creation of an excessive amount of free radicals [3,4]. Additionally, in conditions of intense light, carotenoids can shield chlorophyll from oxidative damage, enabling them to perform a critical defensive function in the struggle against photodegradation [5]. Thus, a great deal of study is still being conducted on plant pigments because of the significance of leaves.

As a distinct and easily distinguishable phenotype, extensive leaf color mutants have been isolated. Depending on the color variation, these mutants can be classified into many categories, such as albino (white), light green, stay-green, dark-green, yellow, yellow-green, red, purple, spotted, or striped [6]. “Zebra” mutants are special color mutants with distinctive yellow-green crossbanding leaves, and several monocotyledonous crop zebra mutants have been described, while the mechanism of zebra leaf formation remains unclear [7]. According to previous research, the zebra leaf phenotype of the rice mutant *zebra2* may be related to photo-oxidative damage in brightly lit conditions. Carotenoids became less abundant once ZEBRA2 was inactivated, and this was followed by an increase in reactive oxygen species (ROS), which led to photoinhibition and photobleaching [8]. Maize *camouflage1* (*cf1*) mutant leaves developed yellow-green sections. The *CF1* gene encoded porphobilinogen deaminase (PBGD), an enzyme involved in the early stages of chlorophyll and heme production. A threshold model that combined photosynthetic cell differentiation as well as the functions of light and ROS homeostasis was considered [9].

During growth and continuing evolution, organisms detect rhythmic changes in their environment and adapt to such periodic variations to establish an internal adaptability mechanism known as the circadian clock [10]. The plant circadian clock system is divided into three parts: the photoreceptor-controlled input pathway, the central oscillator composed of essential elements such as LATE HYPOCOTYL (LHY), CIRCADIAN CLOCK ASSOCIATED 1 (CCA1), and TIMING OF CAB EXPRESSION l (TOC1), and the output pathway involved in plant metabolism, growth, and development [11,12,13]. The *Timeless* (*TIM*) gene was shown to be a component of the central circadian clock associated with the biological cycle rhythm in Drosophila, but up to this point, TIM research had only been conducted in animal models [14]. The plant circadian clock is mostly reflected by the circadian rhythm. Plants adapt to their surroundings by synchronizing their circadian rhythm clock, which promotes the efficiency of resource utilization and energy metabolism, adaptability to the environment, and competitiveness [15].

We recently cloned the *ZmPTOX1* gene by map-based cloning from the maize pigment-deficient mutant *wk3735* and characterized its function during seed development and germination [16]. *ZmPTOX1* encodes a versatile plastoquinol oxidase and is located in plastid membranes. We have validated its important function in maintaining reactive oxygen species (ROS) homeostasis as a crucial component of the electron transport chain (ETC). Additionally, when grown in the field, *wk3735* seedlings exhibit yellow-green crossbanding and demonstrate greater stress sensitivity and less biomass accumulation than wild-type seedlings. While the similar phenotype is ornamental to horticultural plants, we are anxious to determine the mechanism underlying this interesting phenotype. In our study, the relationship between ZmPTOX1 and TIM1 as well as the circadian transcript pattern of *ZmPTOX1* may indicate that the regular phenotype is connected to the circadian rhythm. *ZmPTOX1* was further found to be crucial for plant photosynthesis and stress tolerance using transcriptome analysis in conjunction with the detection of several physiological and biochemical indexes. In this study, we confirmed *ZmPTOX1*’s critical role in maintaining healthy growth in maize seedlings and enhanced the process of zebra leaf phenotypic formation.

## 2. Results

### 2.1. ZmPTOX1 Mutation Produces Yellow-Green Crossbanding and Affects the Photosynthetic Efficiency in Seedlings

We illustrated that *ZmPTOX1* was involved in carotenoid synthesis, and its mutant *wk3735* produced pale-yellow seeds [16]. In contrast to the most common carotenoid-deficient mutant seedlings with smooth pale-yellow leaves, *wk3735* seedlings displayed yellow-green crossbanding leaves after four generations of field observations, and the OE seedlings appeared normally (Figure 1a,b). Interestingly, the crossbanding leaves only appeared before the seven-leaf stage of the seedling, while from the eighth-leaf and later, the crossbanding did not completely disappear, but mixed together and showed light yellow, or even light green, later in the development stages. Crossbanding leaves were also seen in the CRISPR/Cas9 lines, such as *ko#1 (knock out mutant#1)*, in which *ZmPTOX1* was edited, demonstrating that *ZmPTOX1* was responsible for the crossbanding phenotype of seedlings (Figure 1c,d). We also detected that the photosynthesis was affected, as shown by the photosynthetic efficiency parameters of TR (transpiration rate), PR (photosynthetic rate), C-H_2_O/CO_2_ (conductance to H_2_O/CO_2_), and LST (leaf surface temperature); the etiolated sections of *ko#1* seedlings displayed decreased chlorophyll content and photosynthetic capacity, which were compared to the green sections (Figure 1e,f).

### 2.2. The Expression of ZmPTOX1 Is Regulated by the Circadian Clock

As the etiolated sections dispersed with such regularity, we wondered if rhythms had a role in the creation of crossbanding leaves. Six-leaf seedlings of inbred line Zheng58 were cultured in normal conditions for two days under a 16 h light/8 h dark cycle before being moved to constant illumination. Whether under normal conditions or constant light conditions, *ZmPTOX1* transcript levels oscillated, displaying a minor circadian rhythmicity comparable with the pattern of the *circadian clock gene timeless1* (*TIM1*) (Figure 2). *TIM1* was a homolog of *Drosophila TIM*, which was essential for insects’ circadian timekeeping [17]. We accidentally discovered that ZmPTOX1 interacted with TIM1 in the yeast system using Y2H assays, which allowed us to further investigate the relationship between *ZmPTOX1* and circadian rhythm (Figure 3a). The interaction was further identified by using LCI and pull-down assays (Figure 3b,c). It demonstrated that the circadian transcript pattern of *ZmPTOX1* was possibly regulated by its connection to TIM1.

### 2.3. Transcriptome Analysis for OE and Segregated Etiolated and Green Sections of ko#1 Seedlings

To better understand how ZmPTOX1 regulates maize seedling performance, we analyzed the transcriptomes of OE (overexpression) seedlings and the etiolated (muW) and green (muG) regions of *ko#1* seedlings at the 5-leaf stage. Three replicates from each group were included in the nine RNA samples that were extracted. With readings for each sample ranging from 11,001,271 to 27,333,115, transcriptome sequencing provided a total of 207,447,058 clean reads (Table 1). In addition, 90.18–93.11% (mean 91.52%) of the reads were mapped after being aligned to the maize reference genome (Zm-B73-REFERENCE-GRAMENE-4.0), with 62.43–83.11% (mean 78.70%) of those reads being uniquely mapped. FPKM = 1 (Fragments Per Kilobase of exon model per Million mapped fragments) of a gene was regarded to be an expressed gene; there are typically 17,571 (range 17,517–17,678 of three replicates) expressed genes in OE lines, 18,362 (range 18,324–18,388) and 19,038 (range 18,992–19,161), respectively, in muG and muW (Table 1, Figure 4a).

The replicates of the various groups were clearly distinguished in our generated principal component analysis (PCA) plots, indicating the validity of our RNA-Seq data (Figure 4b). When gene expression patterns from several samples were compared using hierarchical cluster analysis, it was discovered that the three biological replicates grouped closely, indicating a high-quality transcriptome method (Figure 4c). In addition, muG’s gene expression patterns exhibited a stronger correlation with OE than with muW (Figure 4c). We also examined the 1239 coDEGs of muG vs. muW and OE vs. muW, and the primary secondary impacts of the mutation in ZmPTOX1 in maize seedlings at the 5-leaf stage were changes in photosynthesis-related activities (Figure 4d,e).

### 2.4. Zebra Leaf Etiolated Sections Exhibit a Marked Defect in the Expression of Genes Involved in the Circadian Rhythm and Rhythmic Stress Response

Compared to the green sections of *ko#1* seedlings, there were 792 downregulated genes and 709 upregulated genes in the etiolated sections (Figure 5a). The two most upregulated genes were *Zm00001d016705* (encodes ATPase inhibitor in mitochondria) and *Zm00001d052040* (encodes cytochrome c oxidase), while *Zm00001d009022* (encodes isoflavone reductase) was the most downregulated gene. These genes are all related to oxidative stress [18,19,20].

Enrichment analysis of 1501 DEGs between muW and muG showed that the enriched cellular component categories were primarily related to the chloroplast and photosystem (Figure 5b; Appendix A). The significantly enriched molecular function categories were associated with photosynthesis component binding, oxidoreductase activity, UDP-glycosyltransferase activity, and channel activity. The enriched biological process categories were related to photosynthesis, the oxidoreduction coenzyme metabolic process, the glycolytic process, and the response to stimulus. The lower starch content also verified the lower photosynthesis efficiency in muW (Figure 5d; Appendix A).

In addition, the item circadian rhythm was enriched to a certain extent, and five DEGs were downregulated in muW (Figure 5c). Furthermore, there were significant differences in the expression levels of numerous crucial genes linked to stress tolerance. *Dehydrin 3* (*DHN3*) belongs to the dehydrin family, whose transcription is inhibited when plants suffer drought and low-temperature stress, and *Ferritin 1* (*FER1*) has been reported to positively regulate maize resistance via a ROS burst [21,22]. When compared to muG, muW had higher transcription levels of *DHN3* and significantly lower levels of *FER1*, coupled with lower levels of H_2_O_2_ (Figure 5e). These findings demonstrated that muW has an impaired mechanism for responding to stimuli. As a result of the DEGs between muW and muG, the GO items “response to light stimulus” and “response to cold” were significantly enriched, which is consistent with the alternating circadian rhythms being predominantly caused by light and cold conditions (Figure 5b). The expression of the GATA transcription factor, a circadian rhythm oscillator that regulates retrograde signals from chloroplasts, was noticeably decreased in muW compared to muG [23,24] (Figure 5e). Combining the connection between ZmPTOX1 and TIM1 with their circadian rhythmicity expression pattern, we hypothesized that alternating circadian rhythms in light and cold stresses may help explain why leaves crossband when *ZmPTOX1* was mutated.

### 2.5. ZmPTOX1’s Expression Level Regulates Seedling Stress Resistance

We analyzed ZmPTOX1 expression in different parts of field-grown seedlings. Despite having a crossband phenotype, there was no difference in ZmPTOX1 transcript levels between the etiolated and green sections of mutated seedlings (Figure 6a,b). To gain more understanding of the molecular lesion that was obviously caused by the *ZmPTOX1* mutation, we analyzed 841 upregulated and 490 downregulated co-DEGs (the DEGs that overlapped between the muG vs. OE and muW vs. OE comparisons) (Figure 6c; Appendix A). According to the KEGG enrichment results, the majority of co-DEGs upregulated in *ko#1* were enriched in the categories of “DNA replication”, “mismatch repair”, “homologous recombination”, and “biosynthesis of amino acids”, among others, suggesting that dysfunction of *ZmPTOX1* caused DNA instability and activated cell repair functions for protection. Contrarily, the downregulated co-DEGs associated with metabolic pathways like “arginine and proline metabolism” and “benzoxazinoid biosynthesis” explicitly revealed that *ZmPTOX1* abnormalities led to weaker environmental adaption and defenses (Figure 6d; Appendix A).

Different from when they were grown in the field, *wk3735* and k*o#1* plants developed into smooth, pale-yellow seedlings in an artificial incubator, further validating the role that alternating circadian rhythm stresses play in crossbanding seedling development (Figure 7a). DAB staining revealed that excess ROS were deposited in the *wk3735*, *ko#1*, and, curiously, OE seedlings (Figure 7b). Antioxidase gene transcription was assessed and found to be considerably higher in OE and slightly but not significantly higher in *wk3735* and *ko#1* (Figure 7c). Additionally, our study on the key chloroplast-encoded genes indicated an upregulation in *wk3735*, *ko#1*, and OE seedlings (Figure 7d). The photosynthesis capacity of *wk3735* and *ko#1* seedlings was suppressed, whereas OE seedlings performed similarly to wild-type seedlings (Figure 7e). We hypothesized that an aberrant amount of *ZmPTOX1* dynamically affected the ETC, causing the expression of photosynthetic components like PSI, the ctyb6f complex, and ribulose-1,5-bisphosphate carboxylase/oxygenase (Rubisco) to increase. The OE seedlings, on the other hand, exhibited normal photosynthesis as proven by their entire ETC and normal green seedling phenotype, whereas the ETC in the mutant etiolated seedlings was deficient in *ZmPTOX1*. Taken together, abnormalities (both mutation and over-expression) in *ZmPTOX1* influenced redox and ETC state, and *wk3735* and *ko#1* plant growth came to harm with lower photosynthesis efficiency, while the OE plant grew normally due to its normal adjustment ability.

## 3. Discussion

### 3.1. ZmPTOX1 Is a Novel Zebra Leaf Phenotypic Control Gene with Pleiotropic Effects

Though easily apparent, leaf color variation is a significant and complex agronomic trait that is influenced by external as well as internal factors. Mutations in genes related to chlorophyll metabolism and chloroplast development are the most frequent cause. Previous studies revealed that the synthesis of chlorophyll necessitates at least 15 kinds of enzymes [25]. Any alteration or reduction in either enzyme’s viability can impact the content of chlorophyll, resulting in variation in the color of plant leaves. An example of this is the maize *ygl-1* mutant, which results from a mutation in the gene encoding a chlorophyll synthase similar to the *Arabidopsis* chlorophyllide-a oxygenase (CAO) [26]. In plants, chloroplasts serve as the main locations for photosynthesis. Numerous investigations have shown that leaf color mutants have a low number of chloroplasts, an aberrant chloroplast structure, and even significant internal deterioration [27,28,29]. Chloroplast formation is catalyzed and driven by numerous enzymes and protein factors, and the loss of either protein’s activity leads to underdeveloped chloroplasts, as is seen in maize mutants with variated leaf colors like *elm2* and *csr1* [30,31]. Aside from that, mutations in the genes responsible for nuclear–cytoplasmic interaction, chloroplast protein transport, heme metabolism, carotenoid metabolism, and abscisic acid metabolism, such as *WHY1*, *CRS1*, *ELM1*, and *Z-ISO*, can also affect the color of leaves [32,33,34].

In the maize database (http://www.maizegdb.org), more than 200 genes or QTL sites related to maize leaf color have currently been identified. However, maize leaf color mutants are primarily trapped in the genetic and simple physiologic analysis stage. In this work, the yellow-green crossbanding on the seedling leaves of the maize mutant *wk3735* was described, and the *ZmPTOX1* mutation is what results in the phenotypic of this leaf color variation (Figure 1). The precise process by which zebra leaves arise in maize is still unknown, as only three mutations linked to the zebra leaf phenotype in maize were thoroughly researched in previous studies. Through this research, we have improved our understanding of the process behind the production of zebra leaves by examining a novel mutant of *wk3735*. *ZmPTOX1*, which is homologous to *AtPTOX* in *Arabidopsis*, participates in the electron transport chain, is in charge of the final stage of electron efflux, and is a crucial enzyme for preserving the homeostasis of the redox state in the plastid [35]. *ZmPTOX1* mutations resulted in multiple types of negative impacts, including decreased carotenoid and chlorophyll content, mesophyll cells, photosynthesis, stress resistance, and antioxidant capacity in the mutant (Figure 1, Figure 6 and Figure 7). The aforementioned performances therefore show that the mutation of *ZmPTOX1* impacts several internal regulatory and metabolic processes and that the intrinsic mechanism of phenotypic formation in mutants is more complex than in most leaf color variation mutants.

### 3.2. The Zebra Leaf Phenotype of wk3735 Is Co-Regulated by Internal Signaling Pathways and External Stimuli

Leaf color mutants, particularly zebra mutants, are a type of phenotype that is generally responsive to environmental factors [36]. The phenotype of zebra leaves will alter as the plant grows, and they are typically most noticeable in the seedling stage. The thylakoid membrane system was shown to be capable of destruction, recovery, and reconstruction when the chloroplast ultrastructure of some rice zebra leaf mutants was observed [37,38]. This allowed the leaves to undergo chlorosis and regreen. The leaves were typically identical to their normal chloroplast structure after regreening [39]. The zebra leaf trait gradually disappeared after the ten-leaf stage in the *zb7* mutant, and its phenotypic alterations were influenced by temperature and light [40]. A maize nonclonal sectoring mutant named *cf1* demonstrated a yellow/green variegated leaf characteristic from leaf emergence to maturity [9]. There are other occasions where *zb9* mutant plants showed conspicuous yellow spots at the three-leaf stage, then gradually formed yellow-green crossbands, which eventually stabilized at the ten-leaf stage and were maintained until plant senescence [41]. Similar to *zb7* and *cf1*, the environment has a significant impact on the phenotype of *wk3735*, which showed crossbanding on its leaves at an early seedling stage when grown in the field and an etiolated seedling when grown in a growth chamber (Figure 1 and Figure 7). The expression profile’s analyses also demonstrated that the *ZmPTOX1* expression level had a significant impact on how responsive plants were to outside stimuli (Figure 6).

Most curiously, there were no yellow-green crossbands on any of the *wk3735* seedlings raised in the growth chamber (Figure 7). According to earlier studies, the phenotype of crop leaves is certainly affected by temperature and light conditions, and a photoperiod is required for the appearance of the yellow stripe on maize *cf1* leaves [9,42,43]. Additionally, the *ZmPTOX1*’s circadian transcript pattern and DEGs between muW and muG related to circadian rhythm and light- and cold-resistance demonstrated that the zebra leaf formation required the circadian rhythm (Figure 2 and Figure 5). We attempted to establish the light and temperature cycle to bring the rhythm in the incubator as similar to nature as possible, but so far we have been unable to get the *wk3735* seedlings grown in the incubator to exhibit the zebra leaf phenotype instead of merely producing etiolated leaves. In previous investigations, an additional noteworthy explanation is that ROS has been linked to the development of the zebra leaf phenotype, such as in rice *zb2* [8]. It is also hypothesized that ROS accumulation above the threshold causes cell damage and results in the formation of the etiolated sector on *cf1* leaves. Since ZmPTOX1 is a key oxidase that regulates the redox state’s homeostasis and is necessary for carotenoid production in plastids, it has a significant impact on the accumulation of ROS and the levels of related scavenging enzymes in materials with abnormal *ZmPTOX1* expression (Figure 7) [35]. Furthermore, Arabidopsis plants with the *AtPTOX* mutation display random white sectors on their leaves, so zebra leaves seem to be a trait specific to monocots [44]. It appears that circadian rhythm, ROS homeostasis, plant characteristics, and other unidentified variables regulate the development of *wk3735* zebra leaves; however, more research is required to identify the precise processes.

## 4. Materials and Methods

### 4.1. Plant Materials and Growth Conditions

This study featured plants with mutator inserted mutant *wk3735* and the transgenic lines *ko#1* and OE. Plants were grown in an artificial incubator with a photoperiod of 16 h: 8 h, light: dark, and a temperature cycle of 25 °C day/16 °C night, and Zheng58 inbred line plants were moved to constant illumination for circadian rhythm analysis on the 10th day. In the field, plant materials were grown in Sanya (N′ 18.247872, E′ 109.508268) in the winter and Hebei (N′ 39.485283, E′ 115.974422) in the summer.

### 4.2. RNA Extraction, qRT-PCR, and RNA Sequencing

The StarSpin Plant RNA Kit (Genstar, Beijing, China) was used to isolate the RNA required for both the transcriptome analysis and qRT-PCR. A StarScript II RT Mix with gDNA Remover kit (GenStar, Beijing, China) was used to generate first-strand cDNAs. A SYBR Green I Kit (GenStar) and a QuantStudio 6 Flex system (ABI, USA) were used for each sample’s qRT-PCR, which was carried out in triplicate. As an internal control (CK), GAPDH was applied. Three separate biological replicates of RNA-seq were performed on nine samples, including the muW and muG sections of the fifth leaf from *ko#1* and the same area of OE seedlings at the V5 stage. Paired-end sequencing using RNA-seq libraries was performed with a read length of 150 bp on the Illumina NovaSeq 6000 platform (Appendix A) (Annoroad Gene Technology, Beijing, China).

### 4.3. Sequence Data Analysis

The clean data were mapped to the B73 reference genome (B73_RefGen_v4) using the program HISAT2 (version 2.2.1) after the adaptors were removed and the reads were trimmed for quality control [45]. The mapping analysis was then performed using SAMtools (version 1.17), and the reads mapped in the annotated gene model were counted using FeatureCounts (version 2.0.1) [46]. DESeq2 (https://bioconductor.org/packages/release/bioc/html/DESeq2.html) in R language was used to find the differentially expressed genes (DEGs). The criteria for a significant difference (adjusted *p*-value 0.05) and an absolute fold-change (FC) ≥ 2 were used to identify the DEGs in seedlings. To calculate gene expression levels, the expected number of fragments per kilobase of transcript sequence per million base pairs sequenced (FPKM) was used. The web-based tools agriGO 2.0 and KOBAS (http://bioinfo.org/kobas) were used to generate combined Gene Ontology (GO) terms and enriched Kyoto Encyclopedia of Genes and Genomes (KEGG) pathways [47,48].

### 4.4. Photosynthesis and Gas Exchange Measurements

Between 11 a.m. and 12 p.m. o’clock in the morning, measurements of photosynthesis and gas exchange were made using an open gas exchange system (LI-6400; LI-COR, Lincoln, NE, USA), as previously reported [49]. The measurement parameters included a relative air humidity of about 35%, a leaf temperature of 32 ± 2 °C, an ambient CO_2_ concentration of about 400 μmol·mol^−1^, and a saturating photosynthetic photon flux density of 2000 μmol·m^−2^·s^−1^.

### 4.5. Measurement of Starch and Superoxide

Samples used for extracting starch and H_2_O_2_ and grinding them into powder in liquid nitrogen were the same as those used for RNA sequencing. A total starch assay kit from Solarbio, Beijing, China, was used to determine the starch concentration. Using 1% 3,3′-diaminobenzidine (DAB) staining for histochemical detection, as previously described, superoxide deposits were observed [50]. The green color of the leaves was removed by boiling them in 95% ethanol for 15 min, after which they were kept in 95% ethanol until it was gone. Using procedures that have been published, H_2_O_2_ content was measured [51].

### 4.6. Assays of Protein–Protein Interactions

For the yeast two-hybrid (Y2H) analysis, *ZmTIM1* ORF and *ZmPTOX1* ORF were cloned into the prey vector pGADT7 and bait vector pGBKT7, respectively. After that, the yeast strain MaV203 was cotransformed with derivatives of pGBKT7 and pGADT7. The transformants were initially raised on a selective medium deficient in Leu and Trp (SD/-Leu/-Trp), and then they were transferred to SD/-Ade/-His/-Leu/-Trp media containing 200 mM 3-amino-1,2,4-triazole (3-AT) as a competitive inhibitor of HIS3, along with X-gal for blue color detection. After being incubated for 72 h at 30 °C, protein interactions were observed.

Luciferase complementation imaging (LCI) assays with pCAMBIA1300-nLUC and pCAMBIA1300-cLUC vectors were carried out to detect protein interactions in tobacco leaves using previously described methods [52]. The N- and C-terminal regions of the *LUC* reporter gene were individually fused with the ORF of each interacting pair. Tobacco leaves were co-infiltrated with the *Agrobacterium* strain GV3101 containing nLUC- and cLUC-derived constructs. Luciferin (100 mM) was applied to the leaf surface and incubated there for 15 min in the dark, and a Fusion FX7 system (Vilber, Marne La Vallée, France) was used to measure firefly luciferase activity two days after infiltration.

His-TIM1 and GST-ZmPTOX1 fusion proteins were generated for in vitro pull-down assays by cloning the corresponding genes into the pET30a and pGEX4T-1 vectors. *Escherichia coli* BL21 cells were employed to express and purify fusion proteins as well as empty tags. We used GST-ZmPTOX1 beads to bind His-TIM1. The pull-down assays were carried out as formerly described [53].

## 5. Conclusions

When we discovered that the carotenoid tangent mutant *wk3735* also displays the phenotype of a typical zebra leaf with low photosynthetic efficiency and defective chlorophyll, we were pleasantly surprised. Since the method of zebra leaf emergence is unclear, we wanted to improve it by researching *wk3735*. The circadian expression pattern of the target gene *ZmPTOX1* and its protein’s interaction with the circadian clock protein TIM1 provided evidence that *ZmPTOX1* is under circadian regulation. According to the analyses of the transcriptome, six genes belonging to the circadian term were among the DEGs between the muG and muW sections, providing additional evidence that the circadian rhythm controls the development of the zebra leaf phenotype. Additionally, *ZmPTOX1* is crucial for the plant stress response, as evidenced by the fact that the DEGs of mutants and OE were significantly enriched in multiple categories linked to stress resistance and over-accumulation of ROS in *wk3735* and *ko#1* leaves. Together, the circadian rhythmicity and stress resistance of *ZmPTOX1* are probably factors in the emergence of *wk3735* zebra leaves.

## Figures and Tables

**Figure 1 plants-13-02346-f001:**
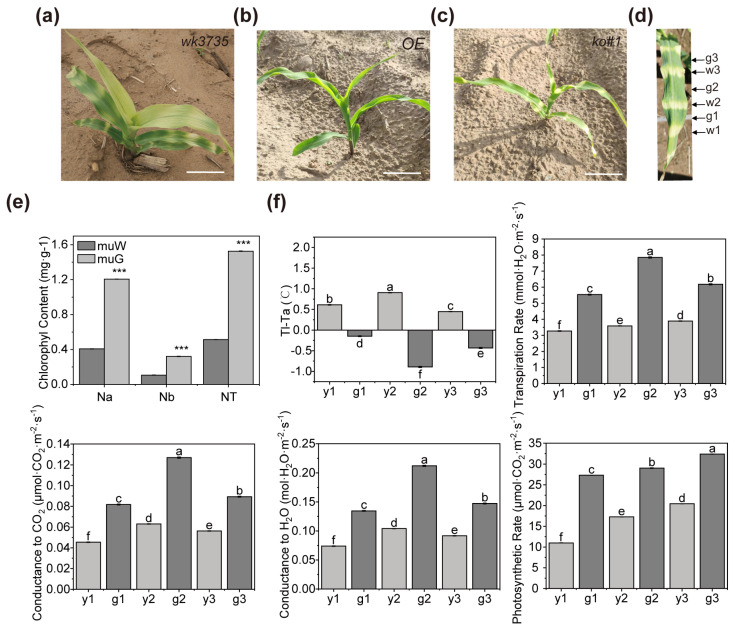
Phenotypic analysis of *wk3735* mutant seedlings. (**a**–**d**) Field-grown mutant seedlings showing apparent crossbanding leaves compared to the normal leaves of OE (V5-stage). w, etiolated region of a mutant seedling; g, green section of a mutant seedling. (**e**) Determination of the chlorophyll content in *wk3735* seedlings. Na, chlorophyll a concentration; Nb, chlorophyll b concentration; NT, total chlorophyll concentration. Values are presented as the means ± SEs (n = 3; ***, *p* < 0.001; Student’s *t* test). (**f**) Photosynthetic parameters were measured using a LI-COR LI-6400 portable photosynthesis system. TI-Ta represents the leaf surface temperature (LST). Significant differences at *p* < 0.05 determined using ANOVA with Duncan’s multiple range test are labeled with different letters.

**Figure 2 plants-13-02346-f002:**
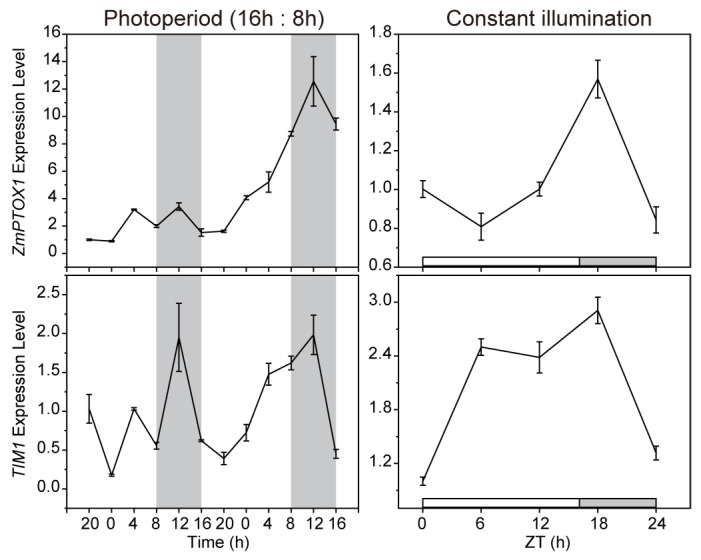
Circadian analysis of *ZmPTOX1* expression along with energy expenditure. Seven-day growth chamber cultivation of seedlings treated with a cycle of 16-h light/8-h dark, then moved to constant illumination. The expression patterns of *ZmPTOX1* and interacted protein timeless1 (*TIM1*) were demonstrated.

**Figure 3 plants-13-02346-f003:**
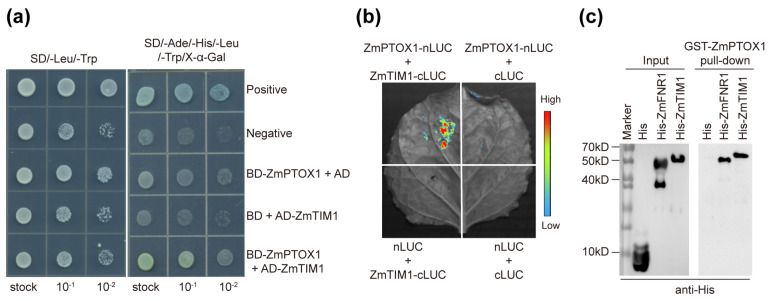
ZmPTOX1 physically interacts with TIM1. (**a**) Y2H assays. AD, GAL4 activation domain; BD, GAL4 DNA-binding domain. Interactions between pGADT7-T and pGBKT7-53 and between pGADT7-T and pGBKT7-Lam were used as positive and negative controls, respectively. To repress autoactivation, 200 mM 3-AT was added to the plate containing SD/-Ade/-His/-Leu/-Trp media. (**b**) LCI assays. The upper left sections show the interaction signal between ZmPTOX1 and ZmFNR1 (**left**) and the self-interaction signal of ZmPTOX1 (**right**); the other sections show the signals of negative controls. The fluorescence signal intensities represent their interaction activities. (**c**) GST pull-down assays. GST-ZmPTOX1 bound to GST beads was incubated with His, His-ZmPTOX1, and His-TIM1, and the elutions were resolved by SDS-PAGE and blotted using anti-His.

**Figure 4 plants-13-02346-f004:**
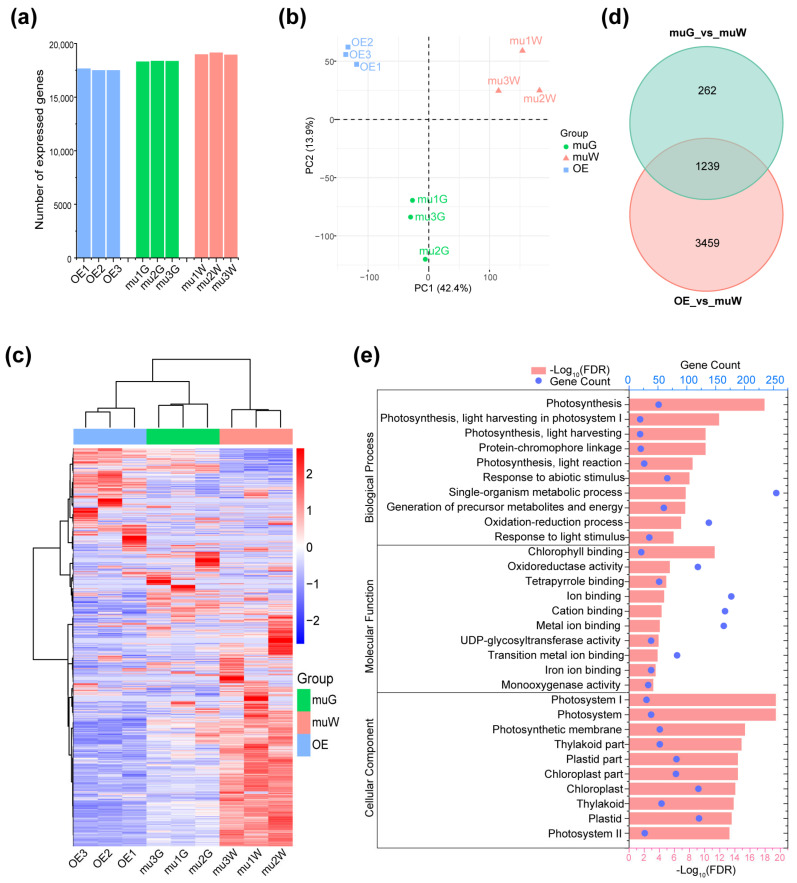
Transcriptome analysis of the mutant and OE seedlings. (**a**) The number of expressed genes identified from etiolated (muW) and green (muG) sections of *ko#1* and OE seedlings. (**b**) Principal component analysis of RNA-Seq data with batch correction. (**c**) Heatmap plotted with FPKM values normalized from 0 to 1 based on each gene. The expression pattern in muG shows closeness to OE rather than muW. (**d**) The number of coDEGs between muG vs. muW and OE vs. muW. (**e**) GO enrichment analysis of above coDEGs.

**Figure 5 plants-13-02346-f005:**
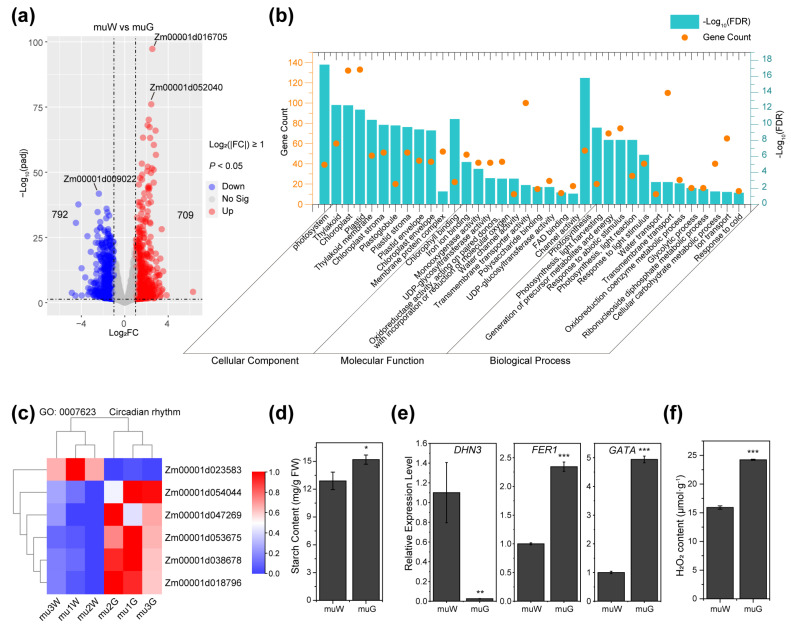
Transcriptome analysis of DEGs between muW and muG and the underlying mechanisms driving the crossbanding of seedlings. (**a**) Volcano plots of DEGs between muW and muG. (**b**) The DEGs were enriched in photosynthesis and in response to stimuli according to the results of a GO enrichment analysis. (**c**) Six circadian genes expressed differently in muW and muG. (**d**) Comparison of starch content between muW and muG. (**e**) Expression levels of *DHN3*, *GATA,* and *FER1* in *ko#1* seedlings, as measured using qRT-PCR. (**f**) Measurements of H_2_O_2_ contents in *ko#1* seedlings. Values are presented as the means ± SEs (n = 3; *, *p* < 0.05; **, *p* < 0.01; ***, *p* < 0.001; Student’s *t* test).

**Figure 6 plants-13-02346-f006:**
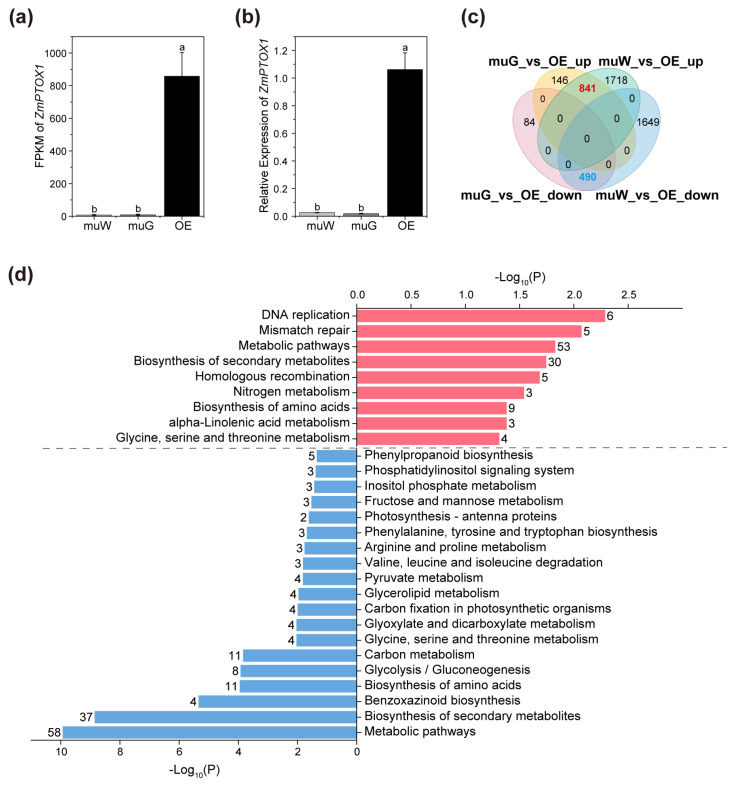
Transcriptomic analysis of the changes induced by substantial differences in *ZmPTOX1* expression. (**a**,**b**) Expression level of ZmPTOX1 detected by RNA-seq (**a**) and qRT-PCR (**b**). (**c**,**d**) Venn diagrams (**c**) and enriched KEGG pathways (**d**) for the overlapping upregulated (upper panel) and downregulated (lower panel) DEGs regulated by *ZmPTOX1*.

**Figure 7 plants-13-02346-f007:**
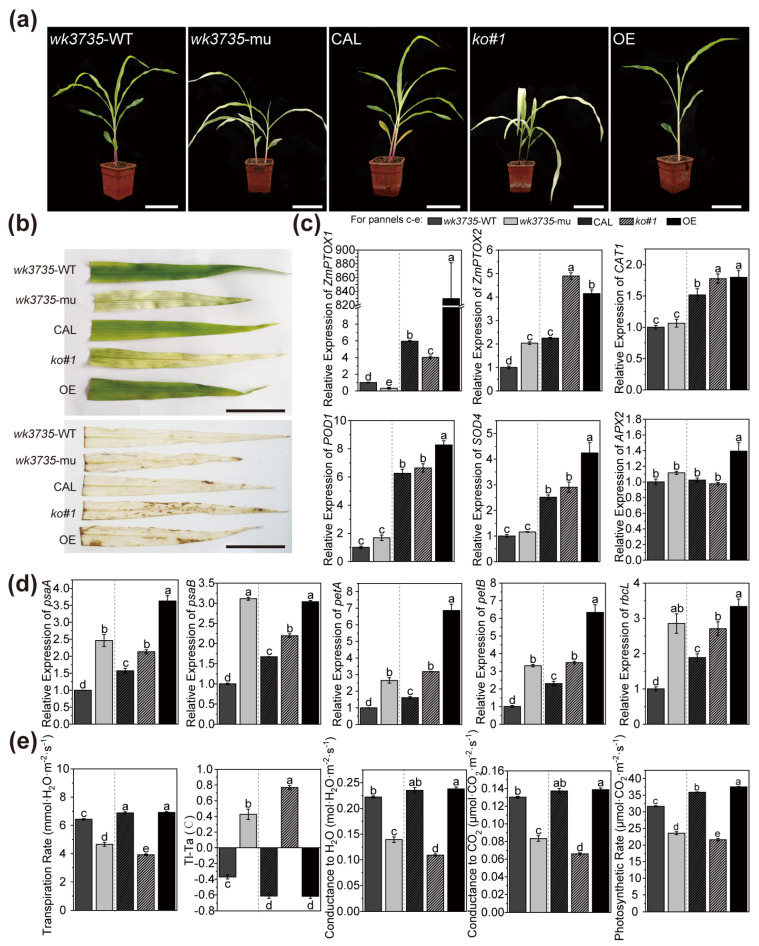
The abnormal ZmPTOX1 quantity fluctuates photosynthesis and ROS homeostasis. (**a**) Seedlings grown in growth chambers for 24 days. Scale bars = 10 cm. (**b**) DAB staining in seedlings. Scale bars = 10 cm. (**c**) Expression level of antioxidases using qRT-PCR. (**d**) Expression levels of photosynthesis components coded by chloroplast genome. (**e**) Photosynthesis parameters measured by LI-COR LI-6400 portable photosynthesis system. Values are means ± SE. Different letters indicate significant differences at *p* < 0.05 determined by ANOVA with Duncan’s multiple range test.

**Table 1 plants-13-02346-t001:** RNA-Seq of 3 replicates of OE, green and white sections of *ko#1*.

Sample Name	Group	Rep.	Total Reads	Rate of Total Mapped Reads (%)	Rate of Uniquely Mapped Reads (%)	Num. of Expressed Genes	Rate of Expressed Genes (%)
OE1	OE	1	26,921,898	98.22	65.09	17,678	38.21
OE2	OE	2	22,588,229	97.67	82.77	17,518	37.86
OE3	OE	3	11,001,271	97.27	82.14	17,517	37.86
mu1G	muG	1	24,837,626	97.54	84.64	18,324	39.6
mu2G	muG	2	22,968,295	97.56	78.88	18,388	39.74
mu3G	muG	3	22,426,755	97.69	84.3	18,375	39.71
mu1W	muW	1	27,333,115	97.47	83.26	18,992	41.05
mu2W	muW	2	25,006,640	97.2	85.83	19,161	41.41
mu3W	muW	3	24,363,229	97.47	83.87	18,963	40.98

## Data Availability

The raw sequence data reported in this paper have been deposited as CRA008125 in the Genome Sequence Archive (Genomics, Proteomics & Bioinformatics 2021) of the National Genomics Data Center (Nucleic Acids Res 2022), China National Center for Bioinformation/Beijing Institute of Genomics, Chinese Academy of Sciences, which is publicly accessible at https://ngdc.cncb.ac.cn/gsa since 1 September 2023.

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
