# Peer review of "Transcriptome Analysis Revealed ZmPTOX1 Is Required for Seedling Development and Stress Tolerance in Maize"

_plants, 2024, doi:10.3390/plants13172346_

Round 1

Reviewer 1 Report

Comments and Suggestions for Authors

I have gone through the manuscript entitled “Transcriptome Analysis Revealed ZmPTOX1 is Required for Seedling Development and Stress Tolerance in Maize″ This study investigated the ZmPTOX1’s critical role in maintaining healthy growth in maize seedlings and enhanced the process of zebra leaf phenotypic formation. The manuscript has a good application value but requires significant improvement in clarity. Many sections are difficult to follow due to complex sentence structures and insufficient background information. In my point of view after the required revision the manuscript could be accepted for publication should the author be prepared to incorporate the major revisions.

Language is just a medium that can convey your message spanning life and important moments of your life to find special findings, so I emphasize the authors to improve the English language and grammar of the article and try to make the explanation as simple as you can to attract new and non-familiar or non-native readers.

The abstract lacks clarity and precision in describing the main findings and significance of the study and contains grammatical errors and difficult phrasing that need correction for clarity. The introduction does not sufficiently review the existing literature or justify the significance of the study. Enhance the introduction by providing a more comprehensive literature review, clearly stating the research gap, and articulating the significance and objectives of the study. Insufficient background on ZmPTOX1 and its known functions and its significance in plant biology. Also lacks a logical flow in the introduction therefore, it is difficult for readers to understand the primary objectives and significance of the research. The results section is not well organized, making it difficult to follow the logical flow of the findings. Some key results are not presented with sufficient data or statistical analysis to support the conclusions. Some results are described without sufficient context or explanation. Discussion seems to be poor, did not give good explanations of the results obtained. I think that it must be really improved. In addition, it should compare and contrast the findings with previous studies to highlight the novelty and significance of the results. Where possible please discuss potential mechanisms behind your observations. You should also expand the links with prior publications in the area, but try to be careful to not over-reach. For the latter, you should highlight potential areas of future study. Literature citation is of great importance, in this article several papers are of older versions, so try to replace recently published articles related to this research. Evaluate the results and link them all in a complimentary manner so that a clear picture draws your efforts for solving the problem in this research.

Comments on the Quality of English Language

Language is just a medium that can convey your message spanning life and important moments of your life to find special findings, so I emphasize the authors to improve the English language and grammar of the article and try to make the explanation as simple as you can to attract new and non-familiar or non-native readers.

Author Response

Dear Reviewer 1,

Thanks very much for your suggestions of this manuscript. Below are the feedbacks/responses to the comments.

Reviewer 1: Comments and Suggestions for Authors

I have gone through the manuscript entitled “Transcriptome Analysis Revealed ZmPTOX1 is Required for Seedling Development and Stress Tolerance in Maize″ This study investigated the ZmPTOX1’s critical role in maintaining healthy growth in maize seedlings and enhanced the process of zebra leaf phenotypic formation. The manuscript has a good application value but requires significant improvement in clarity. Many sections are difficult to follow due to complex sentence structures and insufficient background information. In my point of view after the required revision the manuscript could be accepted for publication should the author be prepared to incorporate the major revisions.

Q: Language is just a medium that can convey your message spanning life and important moments of your life to find special findings, so I emphasize the authors to improve the English language and grammar of the article and try to make the explanation as simple as you can to attract new and non-familiar or non-native readers.

A: Thanks very much for your suggestion. The language and grammar were improved by the professional language organization.

Q: The abstract lacks clarity and precision in describing the main findings and significance of the study and contains grammatical errors and difficult phrasing that need correction for clarity.

A: Thanks for your suggestion. The abstract part was updated, and the grammar was improved. Please see the manuscript.

Q: The introduction does not sufficiently review the existing literature or justify the significance of the study. Enhance the introduction by providing a more comprehensive literature review, clearly stating the research gap, and articulating the significance and objectives of the study. 

A: Thanks very much for your suggestion. We updated the introduction and added more literature related to this context. Please see the highlighted red-colored context in the manuscript.

Q: Insufficient background on ZmPTOX1 and its known functions and its significance in plant biology. Also lacks a logical flow in the introduction therefore, it is difficult for readers to understand the primary objectives and significance of the research. 

A: This part was improved. Please see the highlighted red-colored context in the manuscript.

Q: The results section is not well organized, making it difficult to follow the logical flow of the findings. Some key results are not presented with sufficient data or statistical analysis to support the conclusions. Some results are described without sufficient context or explanation.

A: Thanks very much for your suggestion. We have improved the results since the PTOX1 gene cloning part was published ahead of this work in The Plant Journal. For that part, we focus on gene cloning and gene function in maize kernels, which generated white kernels with improved protein content and a faster gemination rate, while here in this manuscript, we focus more on the function of leaves. For some of the parts, we need to cite the previous work, so we do not describe too much.

Q: Discussion seems to be poor, did not give good explanations of the results obtained. I think that it must be really improved. In addition, it should compare and contrast the findings with previous studies to highlight the novelty and significance of the results. Where possible please discuss potential mechanisms behind your observations. You should also expand the links with prior publications in the area, but try to be careful to not over-reach. For the latter, you should highlight potential areas of future study.

A:  Thanks very much for your suggestion. We have improved the discussion. Please see the highlighted red-colored context in the manuscript.

 Q: Literature citation is of great importance, in this article several papers are of older versions, so try to replace recently published articles related to this research. Evaluate the results and link them all in a complimentary manner so that a clear picture draws your efforts for solving the problem in this research.

A: Thank you very much for your suggestion. We have checked and found that several references are old, and we have updated the versions of these references. In addition, we also made an overall connection with these results to express our efforts to solve the problems in this study. Please see the manuscript for details.

Besides, there are also some revisions that we made in Author list (such as spelling of Xindong Qing, ) affiliates of the authors, funding information, etc.). We also updated the reference list, all the revisions were highlighted as red color in the manuscript.

Reviewer 2 Report

Comments and Suggestions for Authors

The submitted article is interesting, but several elements should be clarified and supplemented before it is published.

1. The introduction lacks a clear justification for the purpose of conducting this research and the choice of maize. Is the problem studied by the authors a common one?

2. The observations made by the authors indicate that the changes in the leaves are caused by the oxidative stress and disorders within the chloroplasts. It would therefore be worth answering an important question, namely what causes oxidative stress and gene mutations in these plants. The circadian cycle itself does not cause these changes since the authors were unable to observe them during the cultivation of plants in a growth chamber.

3. The results were statistically analyzed, but there is no information on this in the materials and methods section. This should be supplemented.

Author Response

Dear reviewer,

Thanks very much for the comments of this manuscript. Below are the feedbacks/responses to the comments.

Reviewer 2: Comments and Suggestions for Authors

The submitted article is interesting, but several elements should be clarified and supplemented before it is published.

Q: 1. The introduction lacks a clear justification for the purpose of conducting this research and the choice of maize. Is the problem studied by the authors a common one?

A: Thanks very much for your suggestion. We updated the introduction, added the literature more related to this context, clearly stated the research gap, and articulated the significance and objectives of the study. Please see the highlighted red-colored context in the manuscript.

Q: 2. The observations made by the authors indicate that the changes in the leaves are caused by the oxidative stress and disorders within the chloroplasts. It would therefore be worth answering an important question, namely what causes oxidative stress and gene mutations in these plants. The circadian cycle itself does not cause these changes since the authors were unable to observe them during the cultivation of plants in a growth chamber.

A: As you can see, there are many moving parts involved in the creation of zebra leaves, including circadian cycles, altered redox states, and unidentified variables that affect phenotypic.

Q: 3. The results were statistically analyzed, but there is no information on this in the materials and methods section. This should be supplemented.

A: Thanks very much for your suggestion. The figure legends provide a detailed description of statistical approaches.

Besides, there are also some revisions that we made in Author list (such as spelling of Xindong Qing, ) affiliates of the authors, funding information, etc.), all the revisions are highlighted as red color in the manuscript.

Round 2

Reviewer 1 Report

Comments and Suggestions for Authors

The author revised his article  accordingly, I recommend it for publication as such in the current form. 

Comments on the Quality of English Language

Minor editing of English language required.